

# Local fishermen's perceptions of the usefulness of artificial reef ecosystem services in Portugal

Jorge Ramos[1,2,3], Pedro G. Lino[4], Amber Himes-Cornell[5,6] and
Miguel N. Santos[4,7]

[1] IPMA—Portuguese Institute for the Ocean and Atmosphere, I.P., Lisbon, Portugal
[2] MARE—Marine and Environmental Sciences Centre, ESTM, Instituto Politécnico de Leiria,
Peniche, Portugal
[3] CinTurs/CIEO—Research Centre for Spatial and Organizational Dynamics, Universidade do
Algarve, Faro, Portugal
[4] IPMA—Portuguese Institute for the Ocean and Atmosphere, I.P., Olhão, Portugal
[5] Ifremer, CNRS, UMR 6308, AMURE, IUEM, University of Brest, Plouzané, France
[6] Fisheries Policy, Economics and Institutions Branch, Fisheries and Aquaculture Policy and
Resources Division, Food and Agriculture Organisation of the United Nations (FAO), Rome,
Italy
[7] ICCAT—International Commission for the Conservation of Atlantic Tunas, Madrid, Spain

Corresponding author
Jorge Ramos, jhramos@ualg.pt

## ABSTRACT

Proponents of artificial reef (AR) deployment are often motivated by the usefulness of such structures. The usefulness of ARs is related to their capability of providing ecosystem services/additional functions. We present two distinct Portuguese AR case studies: (1) The Nazaré reef off the central coast of Portugal and (2) the Oura reef off the Algarve coast. Semi-structured interviews were conducted with local fishermen in the fishing towns of Nazaré and Quarteira pre-and post-AR deployment. The main focus of the interviews was to understand fishermen's perception of AR usefulness (or lack thereof) in terms of nine ecosystem services/additional functions potentially provided by the ARs. We tested the null hypothesis that ARs do not provide additional ecosystem services/additional functions. When queried pre-AR deployment, fishermen indicated that ARs are most likely to provide three ecosystem services: "habitat and refuge," "biodiversity preservation" and "food production." Fishermen had similar perceptions post-deployment. For the Nazaré reef, fishermen tended to have a positive or neutral perception of ecosystem services/additional functions being provided by ARs. For the Oura reef, fishermen tended to have a mostly neutral perception of AR ecosystem services; however, there were also some positive and other negative perceptions. It was difficult for stakeholders to conceptualize some of the ecosystem services/ additional functions provided by ARs prior to actively using them. As a result, some stakeholders changed their perception of the ecosystem services/additional functions after using the structures. These results indicate that stakeholders likely need to perceive ARs as useful in order for them to provide their support for AR installation. Likewise, their support is often needed to justify the use of public funds to install ARs, therefore making it imperative for resource managers to undertake similar interviews with fishermen when considering the use of ARs in other areas.

## INTRODUCTION

Artificial reefs (ARs) have been used to mitigate coastal fishing pressure, enhance the potential of biodiversity, and enhance fisheries catch (*Bohnsack, 1996*; *Leitão et al., 2009*; *Santos, Monteiro & Leitão, 2011*). ARs can function as an effective resource management measure by diverting fishing effort from overexploited fishing areas, or areas which are environmentally vulnerable, to those which are less heavily exploited or less vulnerable (*Bohnsack & Sutherland, 1985*; *Montemayor, 1991*; *Kurien, 2003*). ARs may contribute to reverse fisheries resource depletion (*Watanuky & Gonzales, 2006*). Such reefs can also be utilized for diving activities, diverting divers from sensitive natural reefs to man-made structures (*Wilhelmsson et al., 1998*; *Van Treeck & Schuhmacher, 1999*; *Oh, Ditton & Stoll, 2008*; *Polak & Shashar, 2012*; *Van Treeck & Eisinger, 2012*; *Oliveira, Ramos & Santos, 2015*).

However, in order to be successful, AR projects require many resources (*Whitmarsh et al., 2008*; *Tunca, Miran & Ünal, 2012*). First, AR project proponents need to justify reef advantage and final users need to find reefs useful, otherwise the project may fail (*Edwards & Gomez, 2007*). For a given AR project, proponents need to gather information to justify deployment, such as benefits that have been documented in existing AR projects, in order to advocate for third party acceptance and obtain funding (*Pilkey & Cooper, 2012*; *Lowry et al., 2014*). In this sense, pre- and post-deployment information is important for empowering and providing justification for future AR projects (*Brickhill, Lee & Connolly, 2005*; *Perkol-Finkel & Benayahu, 2005*). Unfortunately, studies involving pre- and post-AR deployment are scarce, particularly those dealing with social or economic perspectives (*Williams, 2006*). Furthermore, although some studies have demonstrated the socio-economic effects of ARs (*Ramos, 2007*; *Seaman, 2007*; *Sutton & Bushnell, 2007*), AR usefulness has been subject to scrutiny (*Pratt, Smokorowski & Muirhead, 2005*; *Brownell, 2011*). For example, *Bortone (2006)* argues that an AR may enhance species richness, but may not ultimately increase human utility.

Interest in the ecosystem services concept has been growing since the late 1990s (*Costanza et al., 1997*, *2014*; *Sterner, 2003*; *Hackett & Dissanayake, 2014*). Many authors have analyzed stakeholder perceptions of the specific ecosystem services provided by various ecosystems (*Lamarque et al., 2011*; *Chan et al., 2012*). Specifically for the case of ARs, studies have shown that stakeholders perceive reef usefulness in various ways depending on whether the ecosystem services that they specifically depend on are enhanced or diminished by the AR (*Sosa-López et al., 2005*; *Chen et al., 2013*). Disagreements between stakeholder groups over the usefulness of ARs can present a significant barrier to obtaining public funds for AR deployment (*Santos, Araújo & Brotto, 2008*). In order to receive public funds, AR project proponents are often required to show evidence of how socially useful the AR will be (*Sawyer, 2001*; *Ramos et al., 2011a*). Given this, one method of supporting the argument for AR deployment is to invest in

**Table 1 Ecosystem services and additional functions potentially provided by artificial reefs (ARs).**

| # | Ecosystem good/service or additional function | Use | Description |
|---|---|---|---|
| 1 | Food production (P) | Current | Direct use (extractive) |
| 2 | Recreational (C) | Current | Direct use (non-extractive) |
| 3 | Biological control (R) | Current | Indirect use |
| 4 | Nutrient cycling (S) | Current | Indirect use |
| 5 | Disturbance regulation (R) | Current | Indirect use |
| 6 | Reuse of obsolete structures (N) | Current | Indirect use |
| 7 | Habitat and refuge (S) | Current | Indirect use and non use |
| 8 | Diversion effect (N) | Current | Indirect use |
| 9 | Biodiversity preservation (S) | Future | Option use and non use |

**Note:**
Ecosystem good or service: Provisioning (P), regulating (R), cultural (C), and supporting (S). Source: Based on Millennium Ecosystem Assessment (MEA) (*World Resources Institute, 2005*). There are two AR effects respondents were asked to consider ("diversion effect "and "reuse of obsolete structures"); however, they are not considered to be ecosystem goods and services. Consequently, they should be considered solely as AR functions. For simplicity, the authors have considered additional functions alongside AR ecosystem services, but they are signed with (N).

multi-functional AR projects, where multiple stakeholder groups are likely to perceive and ultimately directly benefit from the AR (*Ng et al., 2013*).

This paper investigates the varying perceptions of small-scale fishing communities on the usefulness and potential impacts of ARs in two coastal areas in Portugal. The principal research question we address is: How do fishermen perceive usefulness (or lack thereof) of ARs before and after AR deployment? This question is posed for nine ecosystem services and additional functions that were expected to be provided by the ARs. We first describe the ecosystem services and additional functions that ARs are known to provide in general, followed by a description of the AR study areas of interest here. We then present the methodology and questionnaire we used to capture stakeholder perceptions of AR usefulness in both locations, the hypotheses we tested with the questionnaire data, and a summary of the results of the interviews conducted for this study. Lastly, we provide a discussion of the ecosystem services and additional functions perceived by fishermen pre- and post-AR deployment in the two case studies, and justification for using public funds to deploy ARs. We also provide an assessment of the limitations of this study and areas of future research.

## Artificial reef ecosystem services and other functions

Marine ecosystems provide a wide range of services that are fundamental to human well-being and livelihoods, from food production to water filtration, and storm protection (*De Groot et al., 2012*). The total economic value of AR ecosystem services can be divided into the following categories (*Goklany, 2009*; *Huth & Morgan, 2011*; *Johns et al., 2013*): (1) direct use value (extractive and non-extractive use), (2) indirect use value, and (3) option and non-use value. Here, we focus on expected AR ecosystem services and additional functions without considering their inherent economic value (Table 1). The list was not intended to be exhaustive, to be used with fishermen in the two case studies presented here.
## Direct use

### Food production

Food production is an extractive use associated with artisanal and commercial fishing activities that can be modified through AR deployment. For example, *Santos & Monteiro (2007)* investigated the effect of two Algarve AR systems on local fisheries during a 14-year period and found that yields were up to 2.6 times higher at the AR sites than in control areas.

### Recreation

Artificial reefs can provide several opportunities for leisure activities, including surfing (*Ten Voorde, Do Carmo & Neves, 2009*; *Fletcher, Bateman & Emery, 2011*; *Rendle & Rodwell, 2014*), diving (*Ditton et al., 2002*; *Musa et al., 2011*; *Ramos, Oliveira & Santos, 2011b*), spear fishing, and angling from charter boats (*Milon, 1989*; *Chen et al., 2013*). Spear fishing and angling are also extractive and may be categorized as "food production." However, we categorize these activities under "recreation" because the principle aim is for personal consumption rather than for commercial sales.

## Indirect use

### Biological control

Resource managers can introduce ARs to control undesired organisms through predation, herbivory, or any other natural process. In this way, ARs can act as artificial habitats that alter the distribution of species and local biodiversity (*Shipp, 1999*; *Nicoletti et al., 2007*; *Seaman, 2007*). Some species may be lost and others gained (*Jacobus & Webb, 2005*; *Matsuoka, Nakashima & Nagasawa, 2005*). A particular issue related to biological control is the assessment of invader vs. native species colonization rates on ARs (*Smith, 2010*).

### Nutrient cycling

Artificial reef infrastructure provides substrate that contributes to the production of flora and fauna due to high local availability of nutrients in the water column (*Perkol-Finkel & Benayahu, 2005*; *Doyle & Havlick, 2009*; *Levrel, Pioch & Spieler, 2012*). *Falcão et al. (2007)* demonstrated that the presence of ARs may increase the deposition of nutrients derived from higher number of marine organisms that will settle there (i.e., higher carrying capacity). This has a positive impact on the production of fish and other species (*Relini et al., 2007*).

### Disturbance regulation

Concrete ARs are increasingly being used as a shoreline protection measure (*Bleck & Oumeraci, 2001*; *Goudas & Katsiaris, 2003*). ARs can be installed to minimize coastal damage from storms by absorbing wave energy (*Ding et al., 2013*). *Clauss & Habel (2000)* note that ARs may not provide storm protection if they are not strategically located or are too far from the coast. Therefore, if the purpose of an AR is to act as a wave attenuator, they should be deployed close to the shoreline. Conversely, if an AR is meant to increase biological

production, then it should be deployed in areas that are not significantly affected by wave disturbance (*Ten Voorde, Do Carmo & Neves, 2009*; *Morang, Waters & Stauble, 2014*).

### Habitat and refuge

Artificial reefs can be deployed to create habitat and refuges for marine species if adequate depth, temperature, salinity, dissolved oxygen, and the type of materials used are appropriately considered in their construction (*Gallaway et al., 1999*; *Marzinelli et al., 2009*). ARs also provide additional surfaces, holes, and crevices for settlement of sessile organisms (*Clark & Edwards, 1999*; *Moura et al., 2006, 2007*). Such habitat serves both as a source of food for herbivores (*Einbinder et al., 2006*) and as a refuge for many species from predators (*Simon, Pinheiro & Joyeux, 2011*; *Ford & Swearer, 2013*). Habitat complexity increases fish assemblages (*Charbonnel et al., 2002*); therefore, any artificial structure that provides refuge for fish and other marine organisms provide habitat that was unavailable previously (*Busch et al., 2012*).

## Option use

### Biodiversity preservation

Artificial reefs can be used to improve biodiversity in localized areas. *Simberloff & Von Holle (1999)* contend that biologically disturbed areas have fewer species than undisturbed ones, which means for the case of an AR that an increase in biodiversity may be evidence of biological success (*Allemand, Debernardi & Seaman, 2000*; *Lamberti & Zanuttigh, 2005*). ARs can also be used to reverse declines in commercial species. Due to the difficulty in accessing ARs, ARs not only deter fishermen from fishing in the immediate area or over the area (due to the potential for boat and gear damage), but also decrease fishing pressure in the surrounding areas. This is particularly important during early life stages, where increasing the life expectancy of marine organisms living on an AR acts as insurance for larger catches in the future that can be caught in areas adjacent to the ARs (*Goulder & Kennedy, 2011*).

## Additional AR functions

For the purpose of this study, we consider two additional AR functions beyond ecosystem service provision, namely: reuse of obsolete structures and fishing effort diversion. Both of which have been shown to provide important benefits or impacts.

### Reuse of obsolete structures

Artificial reefs are generally made from structures that are no longer in use, such as scrap materials in combination with reinforced concrete (*Bell, Moore & Murphey, 1989*; *Gu, 2005*). However, this has been criticized and considered to be ocean dumping that is being done in the vain of fish habitat enhancement (*MacDonald, Mitsuyasu & Corbin, 1999*; *Chojnacki, 2000*). For example, in the Faro AR complex in the Algarve region of Portugal, an obsolete 35 m iron hull trawler was purposefully sunken in order to provide additional diving spots for recreational use (*Ramos et al., 2006*; *Santos, Monteiro & Leitão, 2011*).

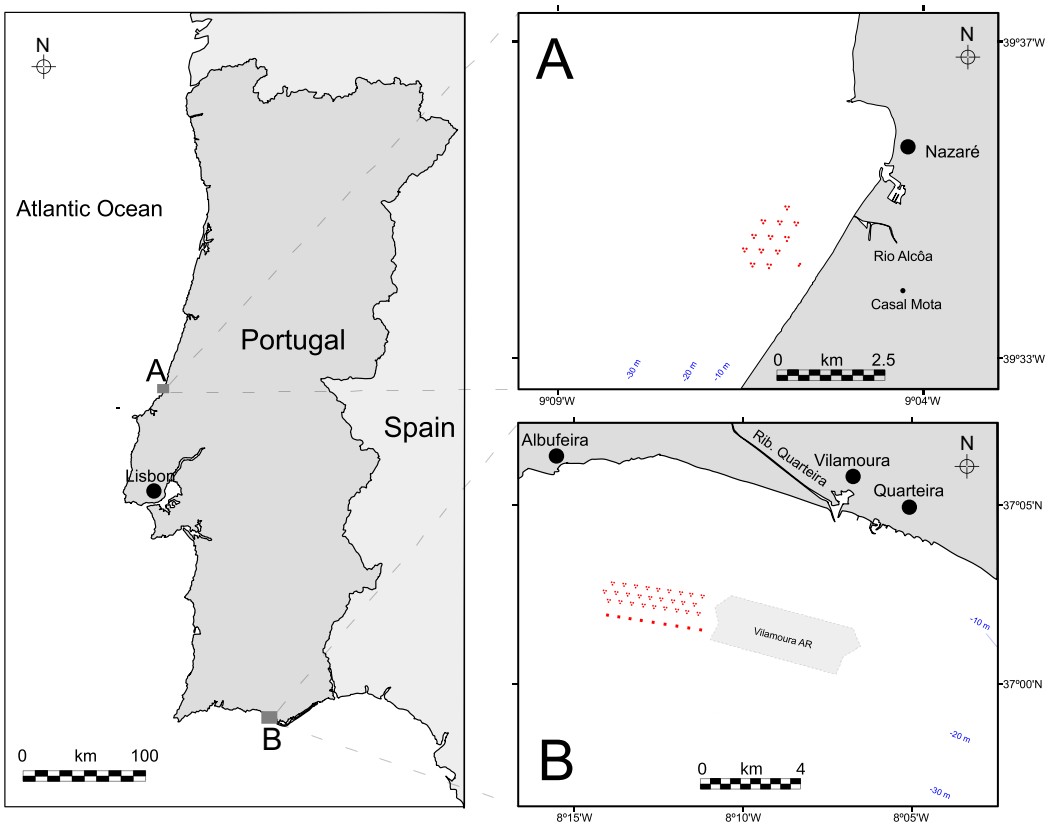

**Figure 1 Case studies location off Portugal.** (A) Nazaré artificial reef (Center region), and (B) Oura artificial reef (Algarve region). Depth is represented by the blue lines and the red shapes show the artificial reef placement.

### *Fishing effort diversion*

Artificial reefs can function as an effective resource management measure to reverse fisheries resources depletion by diverting fishing effort from overexploited fishing areas, or areas which are environmentally vulnerable, to those which are less heavily exploited or less vulnerable (*Bohnsack & Sutherland, 1985*; *Montemayor, 1991*; *Watanuky & Gonzales, 2006*; *Kurien, 2003*). ARs can also be utilized to manage diving activities, diverting divers from sensitive natural reefs to man-made structures (*Wilhelmsson et al., 1998*; *Van Treeck & Schuhmacher, 1999*; *Oh, Ditton & Stoll, 2008*; *Polak & Shashar, 2012*; *Van Treeck & Eisinger, 2012*; *Oliveira, Ramos & Santos, 2015*).

## MATERIALS AND METHODS

### Study area

This study focuses on AR deployment case studies located off the coast of Nazaré (Fig. 1A; central coast of Portugal) and Oura (Fig. 1B; Algarve, southern Portugal) (Fig. 1). In both case studies, the ARs were deployed between 10 and 40 m depth, but in areas with very different bottom topography (Fig. 2).

The two case study AR sites have very different geographic profiles (Fig. 2), and can be summarized as follows.

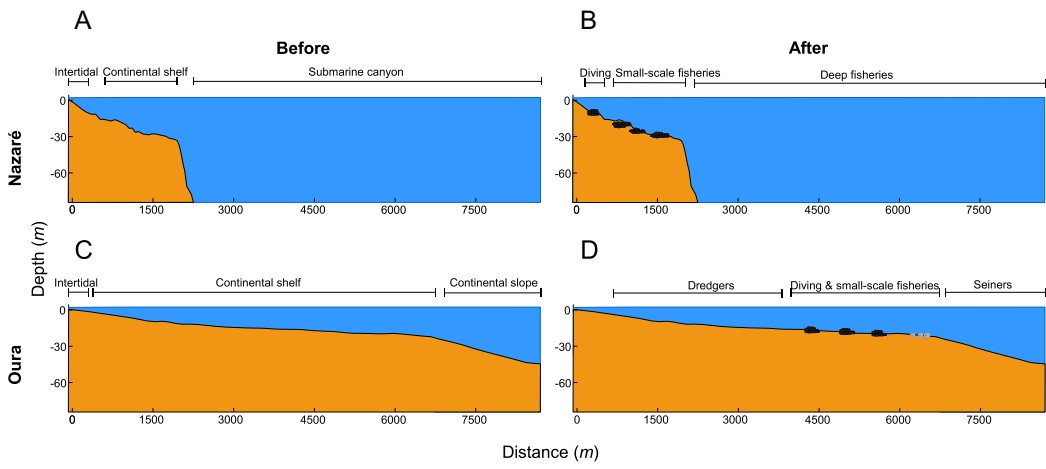

**Figure 2 Artificial reef profile and use (not at scale).** (A and C) represent pre-AR deployment (Nazaré and Oura, respectively), (B and D) represent post-AR deployment (Nazaré and Oura, respectively).

### Nazaré artificial reef

The water off Nazaré is home to a diverse assemblage of fish species, which benefits small-scale fishermen that are able to catch fish species that normally inhabit deep water closer to the coastline. The continental shelf extends out from the coast for approximately 2,000 m and then drops off sharply into a canyon (Figs. 2A and 2B). The central coast of Portugal is subject to harsher weather and more frequent physical disturbance than the southern coast, which in turn affects the ability of coastal human activities to operate (*Tyler et al., 2009*; *Lynn, 2013*). The success of previous AR projects in southern Portugal (*Santos & Monteiro, 2007*) motivated the development of the Nazaré AR project to improve fishermen safety during rougher weather by facilitating the access to a fishing site closer to shore. An AR project was commissioned by the local council in consortium with technical experts from Portuguese fisheries and hydrography research institutes and deployed in 2010. The Nazaré AR covers an area of approximately three km$^2$.

### Oura artificial reef

Adjacent to the Oura coastline, the inner continental shelf extends for over 6,000 m from the coast (Figs. 2C and 2D). In the early 1990s, two pilot ARs were deployed in southern Portugal and showed increased catches (*Santos & Monteiro, 1997*, *1998*). Following the positive results of these pilot ARs in southern Portugal, the Oura AR was deployed during the second phase of a large effort to deploy ARs across the Algarve coast (*Santos & Monteiro, 2007*). The Oura AR was deployed in 2003 next to the Vilamoura AR (deployed in 1998), covering an area of approximately six km$^2$ (*Ramos & Santos, 2015*). The AR was primarily deployed provide a new fishing opportunity to the community of small-scale fishermen in the Algarve.

## Data collection

In order to assess fishermen perceptions of AR usefulness in the two case studies, we developed a questionnaire focused on stakeholder expectations and perceptions of AR

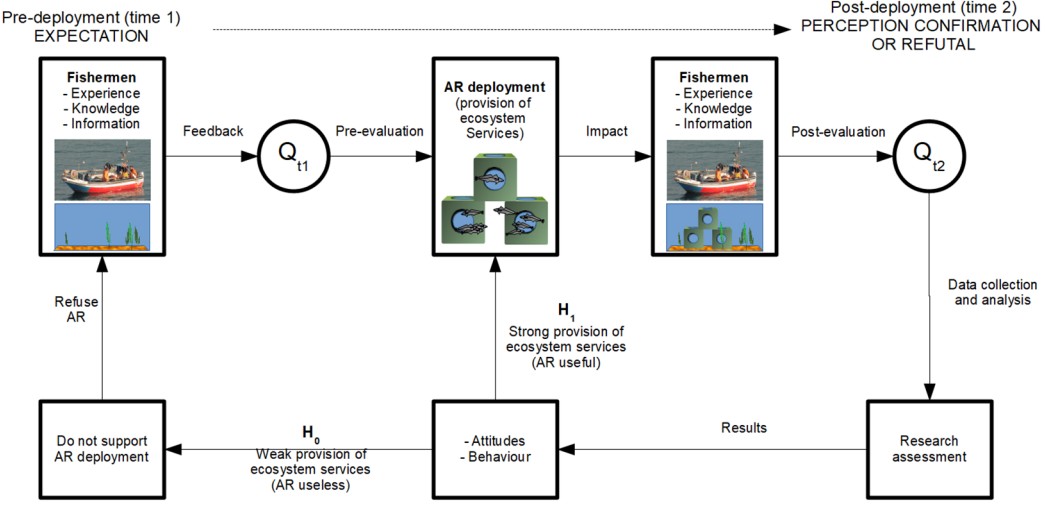

**Figure 3 Study design indicating this study's hypotheses regarding fishermen's expectations and perceptions of ecosystem services and additional functions related to AR deployment.** $Q_{t1}$ represents the pre-deployment questionnaire, whereas $Q_{t2}$ represents the post-deployment questionnaire.

ecosystem services and additional functions (Fig. 3). We deemed it necessary to better understand the advantage that potential AR users expect from the structures both before deployment (pre-deployment), and after the structures had already been deployed (post-deployment). The primary users of these ARs are small-scale fishermen and dive operators; however, the main focus of the AR deployments in both locations was to promote fisheries. Since our main aim was to better understand fishermen's perceptions, diving operators were not considered in this study. Therefore, the potential survey population for this questionnaire included all fishermen who participate in fisheries that are generally prosecuted adjacent to the ARs.

Two questionnaires were used to collect primary data to determine (1) fishermen's expectations for AR benefits pre-deployment and (2) fishermen's perceptions of actual benefits obtained post-AR deployment. These data were then used to test the model's hypotheses (see 'Hypotheses'). Pre- and post-deployment primary data were collected through the implementation of semi-structured questionnaire-based interviews with fishermen in each case study location.

## Pre-AR deployment questionnaire

A pre-AR deployment questionnaire ($Q_{t1}$) was implemented with the aim of better understanding what ecosystem services fishermen believe AR could provide (Table 2). The questionnaire was implemented through face to face interviews in situ at the fishing ports in April 2003 in Quarteira and October 2008 in Nazaré. The questionnaire included demographic questions, but for the purposes of present study, we only present data and analyses related to questions on ecosystem services and additional functions.

Stakeholder viewpoints and preferences can provide important input into the management of marine resources (*Himes, 2007*). Having the above in mind, questions

**Table 2  5-point Likert scale questions posed to respondents.**

| Question | Answer categories | Ecosystem service or additional function |
|---|---|---|
| *Pre-deployment: By deploying an AR in (your specific area), do you think that . . .* | | |
| *Post-deployment: After using an AR in (your specific area), do you think that . . .* | | |
| The production of fish and other seafood is . . . | | Food production (ES) |
| Recreation such as Scuba diving or sea angling is . . . | | Recreation (ES) |
| The absence of plagues or unwanted organisms is . . . | 1. Much less | Biological control (ES) |
| Cleaner waters in the area are . . . | 2. Slightly less | Nutrient cycling (ES) |
| Coast protection against sea storms is . . . | 3. The same | Disturbance regulation (ES) |
| Reuse of scrap, wreck or obsolete structures is . . . | 4. Slightly more | Reuse of obsolete structures (AF) |
| Shelter or refuge for young or vulnerable fish is . . . | 5. Much more | Habitat and refuge (ES) |
| Lack of dredging, trawling or other active gear on the area is . . . | | Diversion effect (AF) |
| The chance of finding many different organisms in the future is . . . | | Biodiversity preservation (ES) |

**Note:**
Right side column in brackets means ecosystem service (ES) or additional function (AF).

were designed to collect fishermen expectations of seven ecosystem services and two additional functions. The interviewer provided each respondent with a brief explanation of each ecosystem service and additional function before beginning the questionnaire. The interviewer then asked the respondent to score their perception of how likely they think the AR will provide each ecosystem service and additional function. Respondents were given a 5-point Likert scale as a means to rate their expectations of future ecosystem service and additional function provision: 1 (much less), 2 (slightly less), 3 (the same), 4 (slightly more), or 5 (much more) (*Allen & Seaman, 2007*; *Dawes, 2008*; *Schmidt et al., 2017*). For example, a score of 4 indicates that the respondent expected that the presence of the AR would provide for slightly more of a given ecosystem service; a score of 2 indicates that the respondent expected that the presence of the AR would provide for slightly less of an ecosystem service. "Don't know" answers were rejected. The questions provoked answers based on the respondent's past experience, knowledge, and information about future expectations on AR.

We defined the survey population by cross-checking the number of vessels found anchored in their respective fishing ports and those recorded in the European fleet register. We compiled secondary data on small-scale fleet sizes in each case study location (*CFR, 2016*). According to the European fleet register (*CFR, 2016*), On December, 31st 2005, there were a total of 87 fishing vessels registered in the European fleet register by fleet category length C2 (6, 10) and C3 (10, 12) in the Nazaré port and 129 fishing vessels in the Quarteira port. Based on this, we determined that an adequate sample size was 20% of the small-scale vessel skippers ($N = 17$ in Nazaré and $N = 26$ in Quarteira). We interviewed a total of 58 skippers (Table 3). A total of 12 responses were not usable because either the skippers were from a non-eligible fishing vessel segment or any other reason.

The interviews were conducted by one researcher and two technicians from the Portuguese Institute for the Ocean and Atmosphere. Respondents were selected by

**Table 3** Number of respondents and usable responses to pre- and post-AR deployment semi-structured questionnaire-based interviews.

| Case study site | Artificial reef phase | | | |
|---|---|---|---|---|
| | Pre-deployment | | Post-deployment | |
| | Participants | Usable responses | Participants | Usable responses |
| Nazaré | 28 | 23 | 24 | 23 |
| Oura | 30 | 23 | 25 | 23 |
| Total respondents | 58 | 46 | 49 | 46 |

contacting fishermen who have demonstrated interest in sharing their opinions about ARs. Fishing community representatives facilitated the process by introducing eligible fishermen to the interviewers. Interviews were conducted in the fisherman's respective port (Quarteira and Nazaré).

## Post-AR deployment questionnaire

A second questionnaire ($Q_{t2}$) was implemented 5 years after AR deployment (May 2008 in Quarteira and September 2015 in Nazaré). The aim of this survey was to gather fishermen's perceptions of the benefits they received from the AR so that the answers provided in the pre-deployment and post-deployment questionnaires could be compared. Our expectation was that respondents had learned from the experience of fishing in the vicinity of the AR. The post-deployment questionnaire included 5-point Likert scale questions similar to the pre-deployment questionnaire, with a focus on the ecosystem services that fishermen believe they have actually benefited from (Table 2). For example, a score of 4 indicates that the respondent realized slightly more of a given ecosystem service after the AR was deployed; a score of 2 indicates that respondent realized slightly less of a given ecosystem service after the AR was deployed. The results were used to better understand fishermen's attitudes toward AR presence and acceptability (or refusal) by the fishing community.

The post-deployment questionnaire followed the same sampling strategy and implementation methodology as described for the pre-deployment questionnaire. On December, 31st 2015, there were a total of 94 fishing vessels registered in the Nazaré port Authority and 118 fishing vessels in the Quarteira port. We interviewed a total of 49 skippers (Table 3). Three responses were not usable because either the skippers were from a non-eligible fishing vessel segment or any other reason.

## Hypotheses

Since we used the 5-point Likert scale data to score the AR effect and the respondents to each questionnaire were not the same pre- and post-deployment, we cannot assume that the population of the ranking scores given by the respondents is normally distributed. Therefore, for each fishing community, we used the Mann–Whitney U-test (also known as Wilcoxon rank-sum test) to test: (1) the null hypothesis that it was equally likely that each of the ecosystem service perception scores given by fishermen would be the same after reef deployment ($H_0$), against (2) the alternative hypothesis that each of the

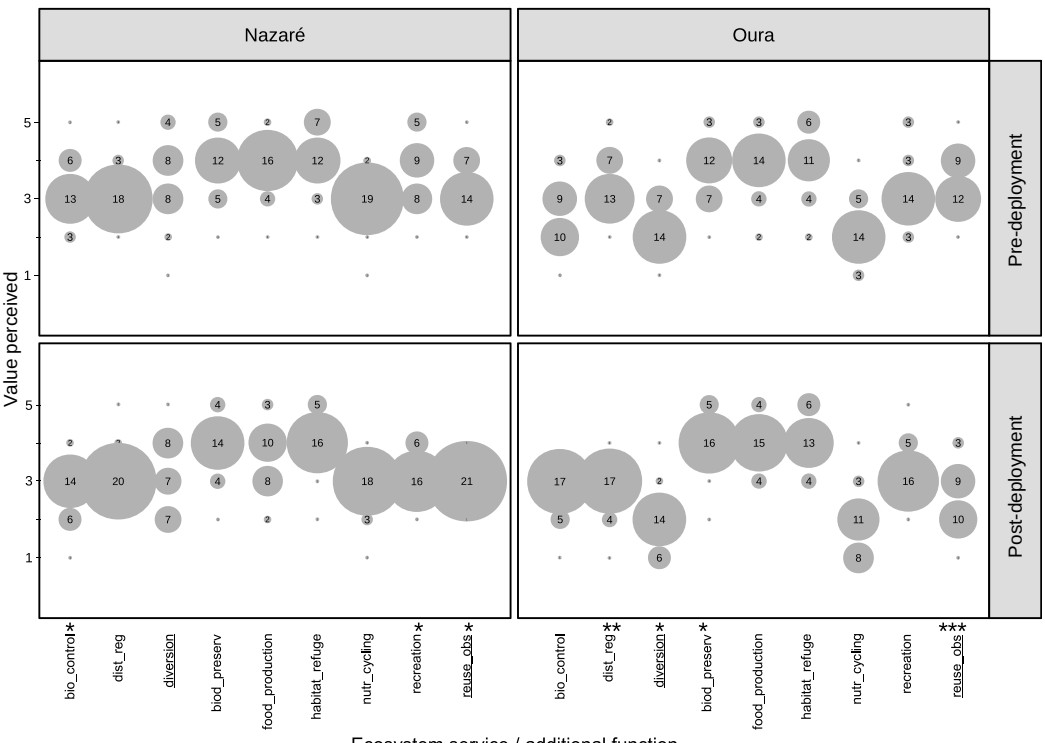

**Figure 4 Bubble plots showing pre- and post-deployment ecosystem service and additional function perceptions found by fishermen on the ARs of Oura and Nazaré (Portugal).** The diameter of each circle and the correspondent figure inside represent the number of respondents' perception of each ecosystem service/additional function. Significance level: $^*p < 0.05$, $^{**}p < 0.005$, and $^{***}p < 0.0005$.

ecosystem service perception scores given by fishermen would differ between the pre- and post-deployment questionnaires ($H_1$). We used R software version 3.1.3 (*R Development Core Team, 2015*) to test both hypotheses.

$$H_0 : \text{Perception of } x_n \text{ pre} - \text{deployment} = \text{Perception of } x_n \text{ post} - \text{deployment} \qquad (1)$$

$$H_1 : \text{Perception of } x_n \text{ pre} - \text{deployment} \neq \text{Perception of } x_n \text{ post} - \text{deployment} \qquad (2)$$

where $X = c(x_1, x_2, x_3, \ldots, x_9)$, $X$ is a set of seven ecosystem services and two additional functions potentially provided by AR and $x_n$ is a given ecosystem service or additional function from $X$.

## RESULTS

In general, for both case study sites and both time periods, respondents indicated that "habitat and refuge" was the most expected (pre-deployment question) or realized (post-deployment question) AR ecosystem service, followed closely by "biodiversity preservation" and "food production" (Fig. 4). Given the small sample size ($n = 23$) and non-normal data sets (ordinal variables), we conducted a non-parametric Wilcoxon test for each case study site individually to compare respondents' perceptions of ecosystem service provision pre- and post-AR deployment (Table 4). The Wilcoxon test indicated
**Table 4 Results of the Wilcoxon rank sum test with continuity correction (two-tailed test) comparing respondents' perceptions of ecosystem service provision pre- and post-AR deployment.**

| Ecosystem service or additional function | Nazaré | | Oura | |
|---|---|---|---|---|
| | W | *p*-value | W | *p*-value |
| 1 Food production (ES) | 308 | 0.2910[n.s.] | 232 | 0.4141[n.s.] |
| 2 Recreation (ES) | 367.5 | 0.0130* | 253.5 | 0.7852[n.s.] |
| 3 Biological control (ES) | 349 | 0.0373* | 235 | 0.4720[n.s.] |
| 4 Nutrient cycling (ES) | 295 | 0.3409[n.s.] | 326 | 0.1391[n.s.] |
| 5 Disturbance regulation (ES) | 265.5 | 0.9867[n.s.] | 382 | 0.0024** |
| 6 Reuse of obsolete structures (AF) | 342 | 0.0231* | 414.5 | 0.0004*** |
| 7 Habitat and refuge (ES) | 268.5 | 0.9295[n.s.] | 243.5 | 0.6222[n.s.] |
| 8 Diversion effect (AF) | 325 | 0.1682[n.s.] | 354.5 | 0.0245* |
| 9 Biodiversity preservation (ES) | 264.5 | 1.0000[n.s.] | 183 | 0.0449* |

Notes:
Left side column in brackets means ecosystem service (ES) or additional function (AF). W is the sum of the ranks of the observations.
Significance level: *n.s.,* non-significant.
\* $p < 0.05$.
\*\* $p < 0.005$.
\*\*\* $p < 0.0005$.

that for most of the ecosystem services, the median Likert scale scores for expected ecosystem services (pre-deployment) were not similar to the ecosystem services that were actually realized (post-deployment) for either case study ($p > 0.05$). However, the median likert scores pre- and post-deployment for the additional AR function of "reuse of obsolete structures" was found to be significant for both locations (Oura: $p < 0.0005$ and Nazaré: $p < 0.05$).

In comparing the pre- and post-deployment responses of Nazaré respondents (Fig. 4), there is slightly higher variability in responses related to "food production" in the post-deployment phase, where many respondents did not perceive that the AR had increased "food production" and the scores for "food production" denoted different catch experiences across respondents. The data distribution is less variable for the ecosystem service "recreation" and the additional function "reuse of obsolete structures," indicating that the combined responses have a higher certainty of the effects derived from obsolete structures after reef deployment.

The Wilcoxon test shows that Nazaré respondents perceived fewer changes in ecosystem services when doing comparisons before and after reef deployment. Most of before and after data presented similar distributions. Significant differences between pre- and post-deployment responses were only found for "recreation," "biological control" and "reuse of obsolete structures" ($p$-value $< 0.05$), for which median responses showed that respondents' expectations pre-deployment for the ecosystem services that the AR would provide were not met.

We found similar pre- and post-deployment responses in Oura (Fig. 4). Respondents tended to score "biodiversity preservation," "food production," and "habitat and refuge" the highest (scored four or above on the 5-point Likert scale) in the pre-deployment

phase, meaning that respondents expected the AR to increase the provision of these ecosystem services. Similar responses were provided in the post-deployment phase.

Oura respondents most often noted that the AR is preserving biodiversity (option use and non-use values). A Wilcoxon test showed that the median score for five of the seven ecosystem services was not significantly different between pre- and post-deployment ($p$-value > 0.05) (Table 4).

By measuring differences in the median perception of ecosystem service and additional function provision pre- and post-AR deployment, we were able to make inferences about respondents' perceived usefulness of the deployed ARs. It was also important to determine if, in general, fishermen perceived the ARs as useful or not.

The results indicate that the null hypothesis ($H_0$) can be accepted regarding six of the ecosystem services and additional functions in the case of the Nazaré AR and five in the case of the Oura AR (Table 4). $H_1$ can be accepted for the option use of "biodiversity preservation" in the case of the Oura AR. The additional function of "diversion effect" was better considered in the Nazaré AR than in the Oura AR, where in the latter case it was considered as the worst additional AR function provided. This may suggest some congestion problems perceived by local fishermen (mostly from the port of Quarteira) that benefited from previous AR experience.

Respondents in both case studies indicated three ecosystem services that triggered beneficial expectations ("food production," "habitat and refuge" and "biodiversity preservation"). A positive trend was only statistically corroborated for the third ecosystem service, thus found for Oura after AR deployment.

## DISCUSSION

### Artificial reef perceptions pre- and post-deployment

Given that benefits of ARs are often only realized in the distant future, it is not easy for fishermen to conceptualize pre-deployment what ecosystem services an AR is likely to provide or increase unless they have previous experience realizing benefits from ARs or other knowledge of ecosystem service benefits associated with ARs. In order to believe in the potential of ARs to provide future benefits, stakeholders will likely need help understanding the benefits of ARs. By providing credible information in advance to support what ecosystem services they can expect from the deployment of an AR, stakeholders may be more likely to support AR deployment (McGlade, 1999). AR proponents need to show fishermen and potential funders how ARs may help to protect fish from intensive fisheries, where they can be deployed, as well as showcase that ARs can provide useful services with some degree of tangibility. AR proponents can also focus on how ARs can be used as an effective fisheries management measure. In the case studies presented here, fishermen who were most likely to use the Oura AR seemed to have benefited from previous experience using ARs that had been deployed nearby. That was not the case for fishermen that were likely to fish around the Nazaré AR, where most had not had the opportunity to benefit from such structures previously. This suggests why comparatively Nazaré skippers were slightly more optimistic than those from Quarteira about the future usefulness of the AR pre-deployment.

Stakeholders may also have improved perceptions of AR usefulness if the benefits are well-explained pre-deployment; however, unless there are known benefits from an existing AR that can be used as an example, it may be difficult for stakeholders and potential funders to fully believe in the potential usefulness of ARs. On the contrary, if stakeholders' expect that an AR will provide positive benefits and then start accruing benefits from the moment they are able to first use it, support for the AR will be high and can be used as an example in the future to advocate for additional AR deployment (*Santos & Monteiro, 2007*; *Kirkbride-Smith, Wheeler & Johnson, 2013*).

However, even when ARs are already put in place and information is provided, additional support may be needed. For example, divers may easily see and attest to increased biodiversity at a dive site, but other stakeholders, such as fishermen, may not directly perceive that they are also benefiting. Ultimately, fishermen must develop some degree of belief that they have realized ecosystem service benefits in order to ultimately believe that the AR is worthwhile. This was exhibited by *Pitcher & Seaman (2000)* as the Petrarch's Principle.

### Reef location profiles and the sense of usefulness

In the Nazaré AR case, fishermen had high expectations for what additional ecosystem services and functions the AR could provide from the beginning. However, after the AR was deployed, fishermen believed that they had realized very few ecosystem services and additional functions compared to their pre-deployment expectations. Overall, the results showed that although Nazaré fishermen were optimistic about the capacity of the AR to provide numerous ecosystem services and additional functions, ultimately the AR did not meet fishermen's expectations. The presence of a canyon nearby may influence fishermen to have different views of AR practicality as fishermen will have better access to both shallow and deeper water fish species.

In the Oura AR case, however, fishermen's expectations before the AR was deployed were on average lower than the Nazaré fishermen expected. Fishermen on average had high expectations for a much smaller set of ecosystem services, namely "biodiversity preservation," "food production" and as a "habitat and refuge." Fishermen's expectations for this smaller set of ecosystem services were ultimately realized in the post-deployment phase. The fishermen's experience of utilizing other ARs nearby may be related to this sense of practicality.

### Cross-checking stakeholders' experience with justifying adequate use of public funds

When public funds are invested in deploying ARs, the direct beneficiaries (e.g., fishermen or other operational stakeholders) create expectations for what benefits they will receive. If those expectations are met post-deployment, AR proponents can argue that deployment was a useful allocation of public money. However, negative experiences where high expectations are not fully met, like the experience of the Nazaré AR, can result in increased scrutiny and can result in reluctance to fund AR projects in the future (*Schuhmann, 2012*).

## Ecosystem services provided by AR and their differentiation

The provision of ecosystem services depends upon complex interactions between organisms and the environment. The need for and importance of individual ecosystem services will depend on the stakeholders that will ultimately be benefiting from them. These factors will therefore affect which ecosystem services are provided, as well as individual stakeholder perceptions of their usefulness. In addition, AR proponents can disseminate information about the likely ecosystem services an AR will provide to potential users. The use of scientific studies, such as the present study, can be used to inform and support future debates on AR usefulness and ultimately improve decision making (*Fisher, Turner & Morling, 2009*). For example, by showing concrete examples where the deployment of ARs has benefited catch potential by protecting juvenile fish from intensive fisheries, fishermen may better understand the potential benefits a new AR may provide them. For less obvious ecosystem services that are not directly related to catch potential, such as "nutrient cycling," it may be more difficult to provide concrete examples of other ecosystem services; however, it is important to develop ways to collect concrete information on the actual benefits that fishermen have realized from ARs for use in future AR proposals.

In the present study, fishermen did not assign high scores to the same AR ecosystem services and additional functions both before and after AR deployment. For the Nazaré AR, there were no nearby ARs that could be pointed to as examples. As a result stakeholder expectations were not based on previous experience. On the other hand, for the Oura AR, the pre-and post-deployment results regarding ecosystem service provision were similar. This was likely due to the fact that most fishermen already had experienced on the use of ARs pre-deployment, and already had a good idea of their usefulness (*Ramos & Santos, 2015*).

Although not tracked, there is a possibility that some of the respondents were queried both before and after deployment. However, since the interviews conducted for this study were anonymous, we cannot confirm which respondents were queried twice. Although this limits further statistical analysis, interviews with the same respondent pool in both the pre- and post-deployment data collections were not possible given the 14 years between studies. Despite this limitation, this study shows that, generally, fishermen across both case studies had a range of expectations of the ecosystem services and benefits that ARs will provide. Likewise, they reported having benefited to some extent from a range of ecosystem services post-deployment. We argue that even if fishermen scored each ecosystem service the same or less post-AR deployment, the ARs are not necessarily deficient in providing ecosystem services. This result likely only indicates that the initial expectations were not completely met and likely unrealistic.

A key objective of the present study was to determine if fishermen generally perceived that ARs do not provide additional ecosystem services and functions (in case of similar scores post-deployment compared to pre-deployment—$H_0$); or if they actually are providing additional ecosystem services and functions (in case of different scores post-deployment compared to pre-deployment—$H_1$). Ultimately, for the two cases presented here, it appears that respondents received positive benefits from ecosystem

services provided by ARs in both study areas, but expectations were not met for most of the AR ecosystems services presented to respondents.

## CONCLUSION

Promoters of AR deployment are often motivated by potential benefits of the structures to human activities. However, apart from a number of studies that have showed positive effects of reef structures in terms of species enhancement, few studies have documented additional ecosystem services that ARs can provide. Even fewer studies have compared pre-deployment perceptions of potential AR ecosystem services and additional functions with those that stakeholders ultimately realize post-deployment. The reason for this may be due to the relatively long delay between when ARs are deployed and when ecosystem services and benefits are realized.

The present study aims to contribute to this knowledge gap by highlighting some of the ecosystem services and functions that ARs have provided to fisheries stakeholders in Portugal. Unless long-term scientific monitoring is planned in advance, it is very unlikely that funds will be available for a pre-deployment assessment and post-deployment monitoring and evaluation such as that presented here. We argue that funds and future research should focus on similar pre- and post-AR deployment case studies in order to continue adding to this body of knowledge. Similarly, AR monitoring and evaluation studies should evaluate potential negative impacts that may be related to AR deployment. Expanded documentation of the effects of AR deployment will provide valuable information that can be used to argue for or against AR deployment in the future.

## ACKNOWLEDGEMENTS

The authors would like to thank the anonymous reviewers of this journal for helpful comments on an early version of this article. The authors would also like to express their gratitude to all respondents from Quarteira and Nazaré for their participation in either the pre- and post-deployment phases of the project. We would also like to thank C. Maurício and J. Delgado as the Nazaré fishing community representatives and H. Rita and Á. Bota from Quarteira fisheries association (Quarpesca), who facilitated the process to collect the data from fishermen.

### Funding

Funding for the Oura AR was provided by a grant from the MARE program, within the project "Implantação e estudo integrado de sistemas recifais." The study also benefited from funding attributed by the PROMAR program (02-PE/2011/GJ) "Elaboração de estudos de caracterização do estado de colonização e impacto socioeconómico do recife artificial da Nazaré." This paper is financed from National Funds provided by FCT—Foundation for Science and Technology through project UID/SOC/04020/2013. The funders had no role in study design, data collection and analysis, decision to publish, or preparation of the manuscript.

## Grant Disclosures

The following grant information was disclosed by the authors:

MARE program, within the project "Implantação e estudo integrado de sistemas recifais."

PROMAR program (02-PE/2011/GJ) "Elaboração de estudos de caracterização do estado de colonização e impacto socioeconómico do recife artificial da Nazaré."

FCT—Foundation for Science and Technology through project UID/SOC/04020/2013.

## Competing Interests

Pedro G. Lino is an Academic Editor for PeerJ.

## Author Contributions

- Jorge Ramos conceived and designed the experiments, performed the experiments, analyzed the data, contributed reagents/materials/analysis tools, prepared figures and/or tables, authored or reviewed drafts of the paper, approved the final draft.
- Pedro G. Lino analyzed the data, contributed reagents/materials/analysis tools, prepared figures and/or tables, approved the final draft.
- Amber Himes-Cornell authored or reviewed drafts of the paper, approved the final draft.
- Miguel N. Santos conceived and designed the experiments, conducted the seminars (pre-deployment).

## Data Availability

The raw data provided is derived from primary data provided in a Supplemental File and was used in a Cran R script.

## Supplemental Information

Supplemental information for this article can be found online at http://dx.doi.org/10.7717/peerj.6206#supplemental-information.

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
