# Peer review of "Local fishermen’s perceptions of the usefulness of artificial reef ecosystem services in Portugal"

_PeerJ, doi:10.7717/peerj.6206_

## Round 0.1 · original submission · Major Revisions

The reviewers agreed that the article has merit, but will require substantial improvement. If you can adequately address the reviewer's issues, we will reconsider the article.

Reviewer 1 ·

Basic reporting

Generally clear and unambiguous, but there were editorial/grammatical errors throughout. Most errors were minor / insignificant (e.g. line 25, a word seems to be missing between ‘deployment’ and ‘semi-structured), but at times, grammatical errors obscured meaning (or at least forced the reader to ‘work hard’ to understand content). Having spent a good part of my life reading material relating to individual and social welfare (utility) functions, I found it particularly hard to understand sections discussing utility. I could not clearly determine if this was an English-Language problem, or if the author(s) were relatively new to the topic, learning it anew and thus struggling a little when explaining ideas. I believe the authors should consider having someone take a more careful editorial look at material before submission.

I think the authors were not as clear as they could be about the difference between the frameworks which economists use to categorise values (the aim there, being to guide selection of appropriate valuation methods, since different types of values, require different valuation methods) and other frameworks that highlight the benefits that ecosystems provide to humans (as per ecosystem services). This is most apparent in Table 1 (also in the associated discussion): some of the ecosystem services that are listed in column 1, might be described by non-market valuation experts in a different way (e.g. in column 3, the authors do not list any ‘non-use’ values – arguably, habitat refuge and biodiversity preservation generate at least some non-use values). So I think the authors could consider either re-visiting the categorisation, develop a more nuanced comparison, or just stick to one. Developing a more nuanced categorisation could be going down a rabbit hole. If using just one framework, I think that Millennium Ecosystem Assessment (MEA) -based frameworks are likely most appropriate (there are many – all slightly different from each other – see for example, the Common International Classification of Ecosystem Services, and more recent material from the IPBES on natures contribution to humans.).

Related to the above - tis very much personal preference, but I tend to think that choice of goods and services for assessment, is a methodological choice – so I would be inclined to move the description of goods and services to later in the paper. My reasoning is as follows: first, you need to work out what you want to assess, then you need to work out how you will conduct the assessment (the before/after interviews, using likert scales), then you need to work out where you are going to undertake the assessment, and who you are going to interview (the sampling). Having collected the data, you are then in a position to analyse it.

While I like the idea of developing a conceptual model linking expectations, knowledge, experience, attitudes and behaviour to perceptions of the ‘value’ of various ecosystem services before and after deployment, this model was neither well explained, nor integrated within the paper itself. I suspect it is used in a larger study, that reports more broadly on the findings in this paper, together with findings that are not discussed here?

Experimental design

I kept wanting to know why/how the authors came up with the list of goods and services to be evaluated? I think the research would have more credibility, if selected goods/services were linked to published categories. That would at least ‘ground’ selection of goods and services in known literature. Whose lead was being followed when devising the list? If some common goods/services were omitted, why? If new ones were added, why?

I thought it a very good idea to do pre and post surveys, but it is not entirely clear if data were collected in such a way as to facilitate true comparisons – e.g. finding out how ‘fisher A’s’ values changed over time. I suspect, from the way results are presented, that this is not the case – hence why the authors are only comparing one sample with another. If it is possible to ‘match’ responses before and after, then I think the authors should consider analysing data differently (looking at changes in values, rather than reporting values before and after separately). If it is not possible to match before and after responses, then authors will have to stick with the sample 1 versus sample 2 comparison, but results are much less interesting and perhaps not strong enough to make a paper on their own. There doesn't really seem to be all that much data to talk about. It is a relatively small sample, before and after, in two case studies, and it is really only Likert responses to questions about several different ecosystem services. I wonder about the idea of trying to add more BREADTH – perhaps including other data (obviously collected in the survey), to help flesh out the conceptual model ...? That could be really interesting.

Figures 4 and 5 could potentially be combined. All that would be required, is the addition of a * (or similar) to Figure 4, near the names of each ecosystem services, for which there were statistically significant differences in distribution of responses on the Likert Scale, between the ‘before’ and ‘after’ survey (I am presuming here, that authors are comparing entire distributions of responses, not just medians).

Validity of the findings

As noted above, it is not entirely clear if data were collected in such a way as to facilitate true comparisons across time – e.g. finding out how ‘fisher A’s’ values changed over time. I suspect, from the way results are presented, that this is not the case – hence why the authors are only comparing one sample with another.

Given the 14 year gap between sample collection, there are likely numerous confounding factors. These were alluded to in the conceptual model - but not systemmatically reconsidered in the conclusion (e.g. noting that some factors would bias responses in this way, others could bias them in another way, with net impacts being ....?). Neither were they seriously considered in the analysis.

I am not 100% convinced that the results definitely 'prove' utility or otherwise (people often respond in patterns to Likert-scale questions, so higher scores may mean higher utility for some goods and services). Moreover, I couldn't work out how to 'ground' the scores --- do they indicate that some Ecosystem services associated with AR's generate more utility than an ice-cream? or a meal? or a house? Is the extra 'utility' associated with the AR's 'worth' more than the cost of constructing them?)

I am also not 100% convinced it is possible to draw inferences about expectations if using two different, unrelated samples, 14 years apart. ........... It might just be a matter of changing language/semantics - my comments are from the economics literature (and utility). There is a fairly large body of economics literature on expectations, measurement of, and interpretation that is not addressed. Perhaps there is a need to discuss that literature (and incorporate insights) or change the terminology of the paper, so that readers are not 'expecting' (:-)) to hear about economics, utility, and expectations.

Additional comments

Tis a good topic - and one about which we know relatively little. Before/after assessments have the potential to contribute enormously to our knowledge. As presented however, I don't feel that the authors are doing their data credit - my personal feeling is that they have, perhaps focused on too small a subset of data from the whole survey. Broadening the variables used in the analysis could add some very interesting additional insights.

Reviewer 2 ·

Basic reporting

It is interesting to compare the impacts of pre and post artificial reef (AR) deployment for different areas (Nazaré reef off the central coast of Portugal and Oura reef off the Algarve coast),different time periods (pre and post) and nine ecosystem services. The literature review is sufficient for the subject matter covered in the article. The manuscript conforms to PeerJ standards and the discipline norm. Figures provided are relevant to the title and the subject matter discussed. English language was clear and correct up to the section 3, but section 4 onward professional levels began to drop. Some examples where the language could be improved include lines 303 onward – the current phrasing makes comprehension difficult in several places.

Experimental design

The experimental design is not an original primary research; however, it was within the scope of the journal. Research question was well defined, relevant & meaningful. It is however, not contributing to an identified knowledge gap.

In line 205, it states, “in order to assess AR utility, we developed a conceptual model of how stakeholder expectations and perceptions of ecosystem services provided by ARs change from pre- to post- deployment (Fig. 3)”. First, this is not a conceptual model of stakeholder expectations and perceptions on ecosystem services. Ecosystem service is a more complex phenomenon, which cannot be captured through a study of perception, measured using a given 5-point likert scales. Experimental design suggested in the paper is not rigorous enough to perform an investigation on ecosystem services to high technical and ethical standards. The type of investigation on revealing perception of stakeholders using likert scale is not providing sufficient details and information to replicate similar studies elsewhere. Authors have cited papers by Allen & Seaman, (2007) and Dawes, (2008) for previous studies which were used 5-point likert scale as a means to rate stakeholder expectations of ecosystem services. To reveal stakeholder expectations of ecosystem services likert scale is sufficient, however, it cannot use for modelling ecosystem services utility suggested in the paper title.

Two references given below have shown analyses that can be used for modelling ecosystem services.
Katja Schmidt, Ariane Walz, Berta Martín-López, René Sachse, (2017)
Testing socio-cultural valuation methods of ecosystem services to explain land use preferences, Ecosystem Services, 26, 270–288 and
M. Potschin-Young, R. Haines-Young, C. Görg, U. Heink, K. Jax, C. Schleyer, (2018)
Understanding the role of conceptual frameworks: Reading the ecosystem service cascade, Ecosystem Services 29 428–440

Validity of the findings

Section 2 of the manuscript provides detailed description of the total economic value of AR ecosystem services which can be divided into three categories, i.e., (a) direct use value (extractive and non-extractive use), (b) indirect use value, and (c) option and non-use value.
Although, the authors have given detailed description of all 9 ecosystem services considered, their impact cannot be assessed. There is no novelty of describing these services, but it is useful if these services could be assessed.
Data collection for the purpose seems robust, statistically sound, & controlled. However, the analysis performed is not novel and it is a very simple approach.
Hypothesis testing of the study is very simple but it was reported wrong. It was stated that a) the null hypothesis that it was equally likely that each of the ecosystem service perception scores given by stakeholders would be the same after reef deployment (H0), against b) the alternative hypothesis that each of the ecosystem service perception scores given by stakeholders would be greater after reef deployment (H1). This is impact true; however, in line 282, it was reported wrong. Instead of “=” sign it was reported as “≥“in null hypothesis.
The conclusion section is well short, linked to original research question but limited to supporting results.

Additional comments

There is no measurement of artificial reef ecosystem services “utility” by local fishermen in Portugal in this study. I commend the authors for their data collection over many years of detailed fieldwork; however, the results are not robust. The manuscript is clearly written in professional way in some parts but not throughout the manuscript. If there is a weakness, it is in the statistical analysis (as I have noted above) which should be improved upon before Acceptance.

Reviewer 3 ·

Basic reporting

The manuscript is not adequately written. There are some grammatical errors, and most importantly there are many sentences whose meaning is not clear. For instance: lines 61-62 – stakeholder perceptions of ecosystem services provision by various ecosystems have been done previously. What is the meaning of this sentence?

There are many more examples of sentences whose meaning is unclear. For example, the text between lines 263-269 is totally confusing: we do not understand if interviewees were asked to rate their perception of ES or to rate the change in their perception before and after deployment of the artificial reefs, which is totally different.

Overall, the manuscript lacks clarity. It would be advisable to have the paper revised by a professional English reviewer.

On the other hand, there is a lack of rigor in the use of concepts. There is a confusion between ecosystem services and (economic) values.
Furthermore, some aspects, such as ‘reuse of obsolete structures’ and ‘diversion effect’ are listed as ecosystem services, but they are not ES. They may be important benefits or impacts of artificial reef structures, but they are not ecosystem services.

There is also a confusion between expectations, perceptions and actual provision of ES – these terms are used throughout the text and sometimes it is not clear to which concept the authors are referring to. (see for example text in lines 243-253).

Experimental design

The main contribution of this paper is the idea of studying users’ perceptions before and after the deployment of artificial reef structures in two different locations. There are not many studies that have managed to have a data set that covers both before and after the project.

However, the research question to be addressed is not clearly defined. It is not clear if the aim of the study was to assess the change in perceptions introduced by the project, or if the idea is to assess the contribution of the artificial reefs to the provision of ES.

As the authors acknowledge, multi-functional artificial reefs provide important benefits to multiple stakeholder groups. However, the study has focused only in one particular group: local fishermen. Although this may be understandable from an operational point of view, at least the authors should acknowledge and discuss this limitation of their study.

The methods are generally well described, but some aspects are not totally clear. The same questionnaire was used in both the before and after interviews? What fraction of respondents was the same in both instances in the two areas?

Perhaps the authors could have compared the results obtained in the interviews with the actual values of fishing yields reported in Santos and Monteiro that they cite in the text.

Validity of the findings

The reported findings are not fully grounded on the data that were collected. Again, there is a confusion between, expectations, perceptions and actual value of ES. The results section focused mostly in the changes in perceptions and dispersion of values. However, in the discussion and conclusions the authors try to conclude about the utility of AR and on the justification of using public funds for AR creation and this analysis is not supported in the data obtained.

Additional comments

The paper addresses a very relevant and timely topic. The main added value is the collection of information before and after the development of the artificial reefs structures. However, the paper needs to be substantially revised before being considered for publication, improving the quality of writing and rigour in the use of concepts and methods, as explained above.

---

## Round 0.2 · Minor Revisions

Good job on the revisions

#

---

## Round 0.3 · accepted · Accept

The article is ready for publication.

#